# INTRIGUING BIAS-VARIANCE TRADEOFF IN DIFFUSION MODELS

## ABSTRACT

Despite the remarkable performance of generative Diffusion Models (DMs), their internal working is still not well understood, which is potentially problematic. This paper focuses on exploring the important notion of *bias-variance tradeoff* in diffusion models. Providing a systematic foundation for this exploration, it establishes that at one extreme, the diffusion models may amplify the inherent bias in the training data, and on the other, they may compromise the presumed privacy of the training samples. Our exploration aligns with the *memorization-generalization* understanding of the generative models, but it also expands further along this spectrum beyond "generalization", revealing the risk of *bias amplification* in deeper models. Our claims are validated both theoretically and empirically.

## 1 INTRODUCTION

Diffusion Models (DMs) show unprecedented capabilities in generating hyper-realistic and diverse synthetic images (Dhariwal & Nichol, 2021; Esser et al., 2021; Podell et al., 2023). Although they have seen rapid empirical progress and widespread adoption in various applications (Liu et al., 2023; Huang et al., 2023; Li et al., 2022; Wu et al., 2023; Watson et al., 2023), their underlying mechanisms and inference dynamics remain comparatively underexplored from a theoretical perspective. This growing gap between practical performance and foundational insight highlights the need for deeper analytical study, which could accelerate the costly training process by diminishing the need for trial and error, and possibly address bias and privacy concerns.

In this work, we explore the traditional concept of bias-variance tradeoff (Geman et al., 1992) in the scope of DMs. Intuitively, our definition of this tradeoff in DMs is roughly analogous to the bias-variance tradeoff in $k$-Nearest Neighbors ($k$-NNs). We conceptualize the training of DMs as *implicit* memorization of the training data, as opposed to *explicit* storing of training samples in $k$-NNs. We show that higher levels of abstraction in the encoding process of DMs are equivalent to larger $k$ values in $k$-NN. So, extremely low abstraction levels lead to "high variance", while excessively high abstraction levels result in "high bias" (see Figure 1). We conceptualize the denoising process of DMs as *memory retrieval* (Hoover et al., 2023), which enables us to set a clear analogy between the way $k$-NNs and DMs treat training data.

Relevant to our exploration, the concepts of *memorization* and *generalization* are also previously studied in the context of DMs (Yoon et al., 2023; Li et al., 2023; 2024; Vastola, 2025). Connecting our analysis to those studies, the established concept of "memorization" falls close to "high variance" as per our analysis, while "generalization" falls somewhere between the two extremes of "high variance" and "high bias". Consequently, our perspective enables more comprehensive insights, naturally revealing that "memorization" and "generalization" are not the ultimate extremes, issuing the caution about going far in the direction of "generalization" that can lead to "high bias" in the models.

Our key contributions are summarized below:

- We systematically explore the bias-variance tradeoff and its implications in the context of DMs.

- We establish that the abstraction level of DM directly impacts bias-variance tradeoff, as extensively abstracted representations lead to exacerbation of bias, while conversely, low abstraction levels can undermine the presumed privacy of DMs.

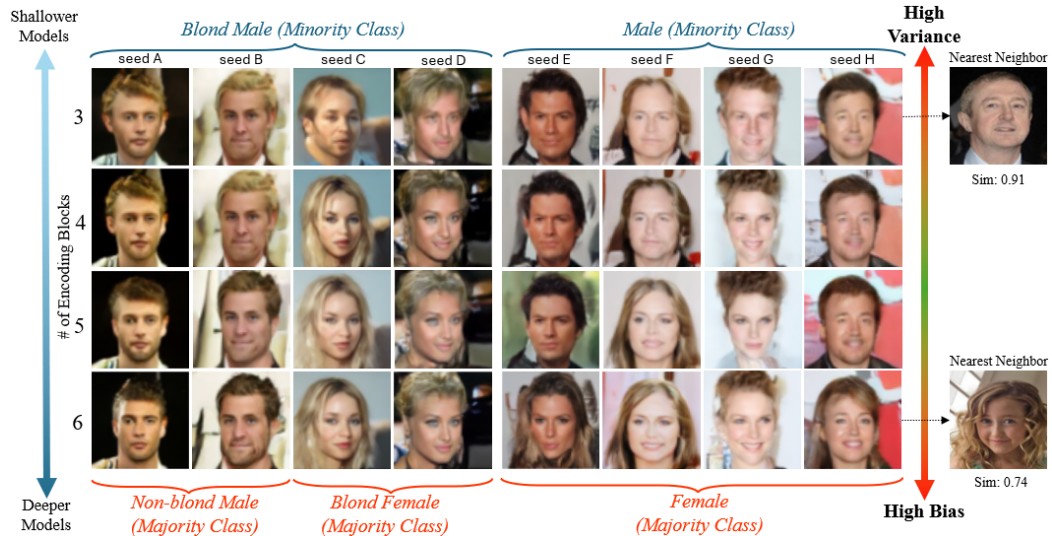

Figure 1: Deeper Diffusion Models (DMs) tend to generate samples of majority classes more compared to shallower models, while shallower models tend to give away information about their training data. Here, unconditional DMs with different depths are trained on CelebA (Liu et al., 2015) that contains "blond male" group and "male" class as minorities (less than 1% and 41.68%, respectively). **High Bias:** Starting from the same seeds on which shallow models generate samples of the minority, deeper models generate samples of the majority. Instead of "blond male", deeper models generate "non-blond male" and "blond female". Also, deeper models generate samples of the majority class "female" instead of "male" . **High Variance:** Shallower models are prone to generate images similar to the training dataset samples. The generated image of the shallow model resembles its nearest neighbor much more compared to the image generated by the deep model.

## 2 DIFFUSION MODELS AS IMPLICIT MEMORY MODULES

To facilitate our "bias-variance tradeoff" explanation, first we need to conceptualize the generative process of DMs as a *"memory recall"* task. As confirmed in (Carlini et al., 2023), DMs can memorize specific training samples and output them during generation. They implicitly store data in latent representations, enabling lossy recall via iterative refinement. This perspective emphasizes the role of the model's learned score function as implicitly encoding a memory retrieval mechanism.

To further explain the memory retrieval behavior, we employ the analogy between energy-based Associative Memories (AMs) (Krotov & Hopfield, 2021) and DMs, inspired by (Hoover et al., 2023). Under this perspective, we view the latent space of DMs as an energy landscape, in which high energy corresponds to noisy signals and low energy represents images resembling the training data distribution. The denoising process starts with a high energy corrupted signal, and by following the direction of negative energy gradients, it eventually ends up in an *attractor*, i.e., a local minimum. In DMs, an attractor can be a *"memory"* or a *"superposition of memories"* (see Figure 2 (A)). The learning phase of DMs can be viewed analogously to traditional memory-based algorithms, such as $k$-NN, in the sense that they both store training data representations. However, it differs to $k$-NN in explicit-implicit dichotomy. In our interpretation of the training process of DMs, training data is *implicitly* memorized by DMs, as opposed to the *explicit* memorization in $k$-NN. Hence, we treat DMs as "Implicit Memory Modules".

## 3 BIAS-VARIANCE TRADEOFF IN DIFFUSION MODELS

Given the analogy between the implicit memorization in DMs and explicit memorization in $k$-NNs, we can gain an insight on the bias-variance tradeoff in DMs. In $k$-NNs, the value of the hyperparameter $k$ determines this tradeoff. Higher value of $k$ leads the query point to attend to a larger number of neighbors, which smooths the decision landscape and reduces sensitivity to individual data points. Similarly, in the context of DMs, an initial corrupted noise (comparable to a query in $k$-NN) attends to a number of implicit memories (comparable to explicit memories in $k$-NN) to generate a sample

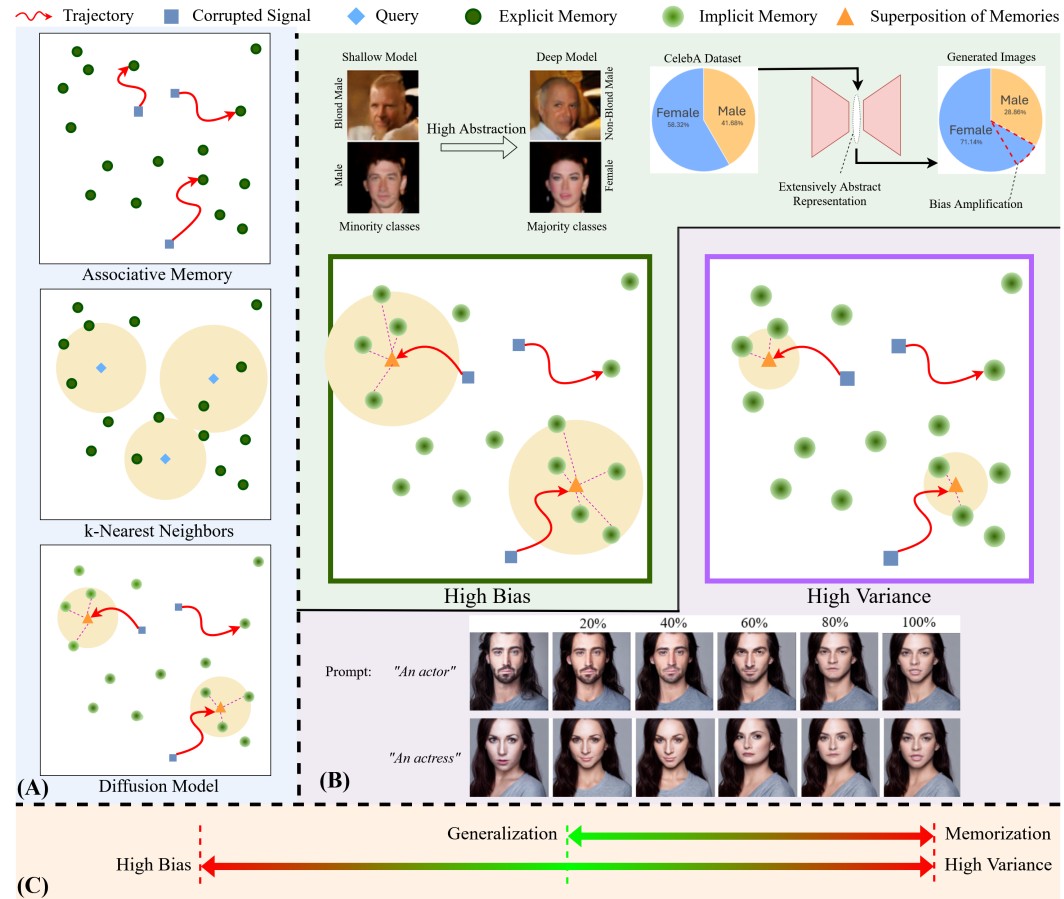

Figure 2: Illustration of the key concepts. **(A)** Comparison of how Associative Memory (AM), $k$-NNs, and DMs store and attend to the training data. In AMs and $k$-NNs, data is stored explicitly, as opposed to implicit storing in DMs. Corrupted signals (noise) in DMs converge to attractors like AMs. However, in DMs, there is no guarantee to converge to a memory, but the attractors are potentially *superpositions of memories*. Like $k$-NNs, DMs attend to a number of training samples during inference. **(B)** *Top:* For smoother energy landscapes, initial corrupted signals tend to attend to a larger number of memories (High Bias), which leads to amplification of the inherent dataset bias. We show actual bias amplification of absolute 12.82% in CelebA (Liu et al., 2015) model. This leads to more frequent generation of the majority class. E.g., "Non-Blond Male" and "Female" are more frequently generated for CelebA by *deeper* models, which naturally inherit higher bias as per our analysis. *Bottom:* For less smooth energy landscapes, initial corrupted signals are likely to attend to a smaller number of memories (High Variance). High variance restricts initial corrupted signals to converge to attractors corresponding to a small number of memories, leading to non-diverse generation, potentially revealing training data that is private. We show examples generated by the prompts "An actor" and "An actress". Starting from the same gaussian seed we gradually push SD V1.5 (Rombach et al., 2022) to a higher variance state during the inference. Both cases eventually converge to the same/similar person. **(C)** Our explanation of "bias-variance tradeoff" for DMs covers the contrasting concepts of "memorization" and "generalization". However, it establishes a more complete perspective beyond "generalization". It captures the risk of falling into "high bias" state by getting too far from "memorization".

resembling the distribution of a class [1] (comparable to categorization to a class in $k$-NN.). Analogous to the higher values of $k$ in $k$-NNs, which lead to the smoothening of the decision landscape, the energy landscape of DMs smoothens with higher levels of abstraction in the DM representations (see Figure 2). Hence, we can expect the bias-variance trade-off in DMs to be linked with the abstraction level of representations.

---

[1]We intentionally treat "concepts" as "classes" to relate our discussion to $k$-NNs.

Table 1: Analogies between $k$-NNs and Diffusion Models (DMs).

|  | $k$-NN | DM |
|---|---|---|
| Storing training data | explicit | implicit |
| Inference initialized with | a query | a corrupted signal |
| Inference process | attends to nearest neighbors | converges to a minimum in nearby memories |
| Smoothing factor | $k$ defined as a hyperparameter | $\tau$ defined by abstraction level of representation |
| Larger smoothing factor | smoothens decision boundaries | smoothens the energy landscape |
| Inference goal | classify the query | refine corrupted signal to resemble a class |

Consider a dataset $D = \{x_i, y_i\}_{i=1}^N \subset \mathbb{R}^d$, where each $x_i$ is a training sample and $y_i$ is its corresponding label, and $Q \in \mathbb{R}^d$ denote a query point (a corrupted signal). In $k$-NN, prediction is based on the average over the $k$ closest training points to $Q$, denoted as $\mathcal{N}_k(Q) \subset D$. The $k$-NN prediction can be written as:

$$\hat{y}(Q) = \frac{1}{k} \sum_{x_i \in \mathcal{N}_k(Q)} y_i. \tag{1}$$

Alternatively, using a soft weighting scheme based on distances, the prediction becomes:

$$\hat{y}_\tau(Q) = \sum_{i=1}^N w_i^{(\tau)}(Q)\, y_i, \quad \text{where} \quad w_i^{(\tau)}(Q) = \frac{\exp\left(-\frac{\|Q - x_i\|^2}{\tau}\right)}{\sum_{j=1}^N \exp\left(-\frac{\|Q - x_j\|^2}{\tau}\right)}. \tag{2}$$

Here, $\tau > 0$ acts as a temperature that controls how broadly the query attends to the dataset: small $\tau$ approximates hard $k$-NN with small $k$, while large $\tau$ induces a smoother, more global averaging.

Now, consider an energy-based interpretation of DMs where the learned memories define a landscape $E : \mathbb{R}^d \to \mathbb{R}$, whose local minima correspond to the data points $x_i$. Given a corrupted input $Q$, the reconstruction follows the gradient flow:

$$\tau \frac{dx}{dt} = -\nabla E(x), \quad x(0) = Q, \tag{3}$$

where $\tau$ controls the rate of change, iteratively mapping $Q$ to a nearby energy minimum. The smoothness of the energy landscape, defined by the number, sharpness, and separation of local minima, determines how many minima influence the trajectory of $Q$. In a rugged landscape, the flow quickly collapses into a nearby basin, analogous to small-$k$ behavior in $k$-NN, where only the most immediate neighbors contribute. In contrast, a smoothed energy landscape causes the flow to be influenced by multiple nearby minima, resembling a $k$-NN with large $k$ aggregating information from many neighbors. Thus, $\tau$ has the same role in DMs as $k$ in $k$-NNs. A comprehensive set of analogies between DMs and $k$-NNs is given in Table 1.

Our definition of "bias-variance tradeoff" aligns with its traditional counterpart, specifically with $k$-NN classification. The bias and the variance of the model $\hat{f}(x)$ is defined by

$$bias(\hat{f}(x_0)) = \mathbb{E}[\hat{f}(x_0)] - f(x_0), \tag{4}$$

and

$$variance(\hat{f}(x_0)) = \mathbb{E}\left[(\hat{f}(x_0) - \mathbb{E}[\hat{f}(x_0)])^2\right], \tag{5}$$

where specifically in our definition of bias-variance in diffusion models, $f(x_0)$ is the class of the training image that is encoded to $x_0$ and $\hat{f}(x_0)$ is the class of the generated image decoded from $x_0$.

### 3.1 IMPLICATIONS OF LATENT SPACE ABSTRACTION LEVEL

Here, we examine how the energy landscape corresponding to DMs' learned memories gets affected by the abstraction level of their latent representation.

As formally depicted in Theorem 1, increasing the level of abstraction in latent representations induces smoother energy landscapes. To support this claim, we analyze the local and global smoothness of the energy function by examining two key quantities: the Hessian norm, which captures local curvature, and the Lipschitz constant of the gradient field, which reflects how rapidly gradients can vary across the space. We show that both the Hessian norm and the Lipschitz constant decrease with higher levels of abstraction.

**Theorem 1** (Higher abstraction in representations induces smoother energy landscapes). *Let* $E :$ $\mathcal{H} \to \mathbb{R}$ *be an energy function defined over a space of representations* $\mathcal{H}$*,* $\phi^{(a)} : \mathcal{H} \to Z^{(a)}$ *be a hierarchy of abstraction maps indexed by level* $a$*, where each* $\phi^{(a)}$ *is a smooth, contractive transformation, and* $Z^{(a)}$ *be a vector space of latent representations after some level of abstraction. Define the energy at abstraction level* $a$ *as*

$$E := E \circ (\phi^{(a)})^{-1} : Z^{(a)} \to \mathbb{R}. \tag{6}$$

*Then the energy landscapes* $\{E^a\}_a$ *become progressively smoother with increasing* $a$ *in the following sense:*

$$\left\| \nabla^2 E^{(a+1)} \right\| < \left\| \nabla^2 E^{(a)} \right\|, \quad \text{and} \quad L^{(a+1)} < L^{(a)}, \tag{7}$$

*where* $\left\| \nabla^2 E^{(a)} \right\|$ *denotes the Hessian norm (curvature), and* $L^{(a)}$ *is the Lipschitz constant of the gradient* $\nabla E^{(a)}$*.*

The complete proof of Theorem 1 is given in the Appendix. As a result of smoothing energy landscapes, some of the nearby minima will be merged into a wider yet shallower minimum basin, which is defined in Definition 1.

**Definition 1** (Merged Local Minimum). *Let* $E : \mathbb{R}^d \to \mathbb{R}$ *be an energy function corresponding to a latent representation. If the energy landscape becomes smoother, then a set of nearby local minima* $\{x_i^*\}_{i=1}^n$ *of* $E$ *may merge into a single, wider and shallower local minimum* $\tilde{x}^*$ *of the smoothed energy function* $\tilde{E}$*. Formally, for some small* $\varepsilon > 0$*,*

$$\|\phi(x_i^*) - \tilde{x}^*\| \leq \varepsilon, \quad \forall i = 1, \dots, n,$$

*where* $\phi$ *is an abstraction (encoding) map. Hence,* $\tilde{x}^*$ *is a representative of the local minima in* $E$ *under the abstraction i.e. in* $\tilde{E}$*. We refer to such a minimum* $\tilde{x}^*$ *as a "Merged Local Minimum".*

The local minima that form a merged local minimum can potentially be from different classes. The question that arises here is what happens if the high-energy corrupted signal ends up in such a new merged minimum. In other words, what is the probability of generating each of the classes residing in the merged minimum? Answering this, we show in Theorem 2 that the generation gets heavily biased in favor of the majority class.

**Theorem 2** (Merged local minima are biased representatives). *Let* $E : \mathbb{R}^d \to \mathbb{R}$ *be an energy function, and let* $\{x_i^*\}_{i=1}^n \subset \mathbb{R}^d$ *be local minima of* $E$*, each associated with a class label* $y_i \in \mathcal{Y}$*. Define the convex hull of these minima as:*

$$H_C(\{x_i^*\}_{i=1}^n) := \left\{ \sum_{i=1}^n \alpha_i x_i^* \mid \alpha_i \geq 0, \sum_{i=1}^n \alpha_i = 1 \right\}. \tag{8}$$

*Assume that an abstracted or smoothed version of the energy function* $\tilde{E}$ *satisfies:*

$$\tilde{E}(x) \approx E(x) \quad \text{for} \quad x \in H_C(\{x_i^*\}), \tag{9}$$

*and*

$$\tilde{x}^* := \arg \min_{x \in H_C(\{x_i^*\}_{i=1}^n)} \tilde{E}(x), \tag{10}$$

*where* $\tilde{x}^*$ *is a global minimum of* $\tilde{E}$ *restricted to the convex hull. Define the empirical class distribution among the constituents as*

$$P_n(c) := \frac{1}{n} \sum_{i=1}^n I[y_i = c], \quad \text{for each} \quad c \in \mathcal{Y}, \tag{11}$$

*and let* $c_{maj} := \arg \max_{c \in \mathcal{Y}} P_n(c)$ *denote the majority class. Then the classification of the merged minimum* $\tilde{x}^*$ *is highly biased toward* $c_{maj}$*, i.e.,*

$$\mathbb{P}\left[f(\tilde{x}^*) = c_{maj}\right] \gg \mathbb{P}\left[f(\tilde{x}^*) = c\right], \quad \forall c \in \mathcal{Y} \setminus \{c_{maj}\}, \tag{12}$$

*where* $f : \mathbb{R}^d \to \mathcal{Y}$ *is a labeling function such that* $f(x_i^*) = y_i$*.*

Figure 3: (Illustration of bias amplification.) Five different energy landscapes containing minima for two classes: "red" and "blue" with corresponding proportions of 50%-50%, 60%-40%, 70%-30%, 80%-20%, and 90%-10% are shown (left column). Each of the landscapes gets gradually more abstract, and the class of the minima is decided based on Theorem 2 by getting a local average over classes using a 2x2 window (random in case of tie in the average). The proportion of minima for the 50%-50% case remains almost the same for every level of abstraction (see first row). However, for biased cases, the bias gets amplified, and the amplification rate gets exacerbated for higher levels of initial bias (see rows 2 to 5). The percentage of the "red" (majority) class is given above each landscape.

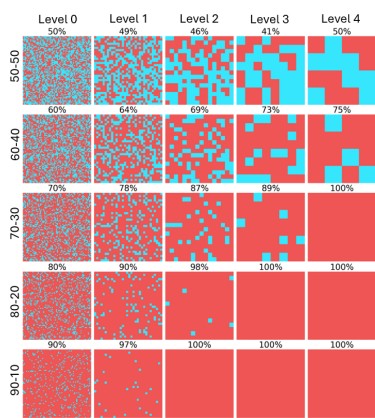

*Proof.* Let $\tilde{x}^*$ be a merged local minimum basin formed by smoothing the energy landscape containing $n$ local minima, with the number of local minima corresponding to classes $A$ and $B$ being $p$ and $q$, respectively. The initial probability of convergence to classes $A$ and $B$ before smoothing is

$$\mathbb{P}_{\text{init}}(A) = \frac{p}{n}, \quad \mathbb{P}_{\text{init}}(B) = \frac{q}{n}. \tag{13}$$

The initial odds ratio of class $A$ to class $B$ is given by

$$\lambda_{\text{init}}(A) = \frac{p}{q}. \tag{14}$$

After merging the minima, the generated image will belong to class $A$, class $B$, or a mixture of both. However, as long as classes $A$ and $B$ are distinguishable and the model is well-trained, the generation of a mixture of classes is not possible. Therefore, the generated image must either be from class $A$ or from class $B$.

To avoid generating a mixture, the model must select all exclusive semantic features from one class. Thus, to generate an image from class $A$, all exclusive features must be selected from class $A$, each with an odds ratio of $\frac{p}{q}$. Let $S$ be the number of exclusive semantic features. The odds ratio of generating an image from class $A$ to class $B$ after smoothing is

$$\lambda_{\text{smooth}}(A) = \left(\frac{p}{q}\right)^S. \tag{15}$$

If $p > q$, for $S > 1$ we have

$$\lambda_{\text{smooth}}(A) = \left(\frac{p}{q}\right)^S > \frac{p}{q} = \lambda_{\text{init}}(A), \tag{16}$$

and for a sufficiently large $S$,

$$\lambda_{\text{smooth}}(A) \gg \lambda_{\text{init}}(A). \tag{17}$$

So,

$$\mathbb{P}_{\text{smooth}}(A) \gg \mathbb{P}_{\text{smooth}}(B), \tag{18}$$

where $\mathbb{P}_{\text{smooth}}(A)$ and $\mathbb{P}_{\text{smooth}}(B)$ are probabilities of convergence to classes A and B after smoothing, respectively. $\square$

Figure 3 illustrates how biased representatives exacerbate bias. It shows five energy landscapes with different proportions of classes, getting gradually more abstract (left to right). Based on Theorem 2, we represent nearby classes as the majority class (random in case of a tie) in the more abstract representations. It can be seen that while class proportions remain almost identical in the balanced case, the majority class dominates with higher levels of abstraction in unbalanced cases. Also, the stronger the initial bias, the greater the bias amplification induced by representation abstraction.

### 3.2 LATENT SPACE ABSTRACTION LEVEL INFLUENCES BIAS-VARIANCE TRADEOFF IN DIFFUSION MODELS

Based on Theorem 1 and Theorem 2, we claim that the bias-variance tradeoff in diffusion models is directly influenced by the level of abstraction of the latent space representation.

Figure 4: % of generated images from the minority class, and the average LPIPS metric to measure their diversity. Five DMs with 2 to 6 encoding blocks each trained on four custom MNIST datasets contained only "6"s and "9"s with biased proportions (see legend). Deeper models, that make more abstract representations, generate more diverse images (higher average LPIPS) and amplify the bias. "High variance" and "high bias" states can be identified on "shallow" and "deep" models, respectively.

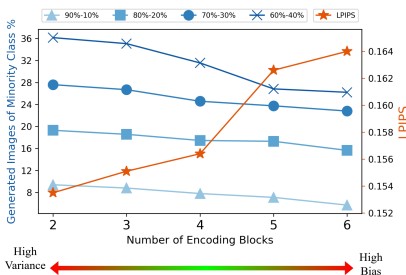

Table 2: Percentage of images generated by unconditional DMs trained on CelebA. Higher abstraction of representations that is achieved by a larger number of encoding blocks leads to amplification of bias, as well as more diversity in generating images.

| | Dataset Proportions | # of Encoding Blocks | | | | | |
|---|---|---|---|---|---|---|---|
| | | 2 | 2 (large) | 3 | 4 | 5 | 6 |
| Male | 41.68% | 40.33% | 40.85% | 39.18% | 37.22% | 30.36% | 28.85% |
| Female | 58.32% | 59.67% | 59.15% | 60.82% | 62.78% | 69.64% | 71.15% |
| Average LPIPS | | 22.91 | 23.26 | 23.17 | 25.65 | 28.03 | 29.12 |

**High abstraction levels of representations lead to "high bias".** Based on Theorem 1, highly abstracted representation makes the energy landscape corresponding to the DM's learned memories smoother, and this smoothness results in the emergence of merged minima basins. Theorem 2 establishes that converging to those merged minima basins amplifies the probability of generating majority class samples considerably. Consequently, the generated images may belong to the majority class with a higher probability than the original proportion of that class. That is, the bias gets amplified. **Low abstraction levels of representations lead to "high variance".** Low abstraction levels lead to uneven energy landscapes with numerous individual local minima, which are representatives of less aggregated and more individual data points. The convergence behavior of the corrupted signal is very sensitive to the presence or absence of individual training samples in this case. As a result, this will be the state of "high variance".

### 3.3 "High Bias State" Leads to Amplification of the Inherent Bias in Datasets

To verify the established relationship between high levels of abstraction and "high bias", we designed and conducted a series of experiments on MNIST and CelebA datasets to further examine this connection.

First, we trained a series of unconditional diffusion models on custom MNIST datasets containing only digits "6" and "9" with different proportions. The base case is a diffusion model with two encoding blocks trained on a biased dataset containing 90% samples of class "6" and 10% samples of class "9". We further trained increasingly deeper models on the same biased dataset. The models consist of three, four, five, and six encoding blocks. Interestingly, we observed that while the percentage of generated images for the base model almost aligns with the proportions of the training dataset, deeper models; which leverage increasingly more abstract representations, tend to amplify the bias. The same experiments are also conducted on datasets consisting 80-20, 70-30, and 60-40 percent of "6"s and "9"s, respectively, and the results were consistent. We also used Learned Perceptual Image Patch Similarity (LPIPS) metric Zhang et al. (2018) to measure the diversity of generated images. For each setting, we generated 1000 images and calculated the LPIPS metric for each pair. Eventually, the average LPIPS is reported, which showed higher diversity for deeper architectures (see Figure 4).

To validate our observation further, we trained a series of increasingly deeper DMs on CelebA dataset Liu et al. (2015) and generated 10,000 images. As presented in Table 2, the proportions of generated images across the two classes of "Male", "Female" support our claims. Although the proportion of "Female" samples to "Male" samples is $58.32\%$ to $41.68\%$ in the training data, the bias amplifies as the number of encoding blocks increases. Skeptically, this observation could be caused by over-parameterization. Hence, we repeated the experiment on a large 2-encoding-block model that

Figure 5: 10 unconditional consecutive generations of two DMs trained on CelebA, starting from the same seeds: one with 6-encoding blocks and one with 4-encoding blocks. The deeper model tends to follow the biased data distribution in CelebA. While most outputs are visually similar, two highlighted cases differ: the 6-encoding block model favors the majority classes, producing non-blond males over blond males (40.81 % vs. 0.86 %) and females over males (58.32 % vs. 41.68 %).

Table 3: Average cosine similarity in Inception feature space and average Euclidean distance in pixel space between generated images and their corresponding $k$ nearest neighbors in the CelebA training dataset. The generated images from shallower models consistently show higher cosine similarity and lower Euclidean distance in the Inception feature and pixel spaces, respectively, indicating their greater tendency to expose training data information.

| # of Encoding Blocks | Avg. cosSim in Inception Feature Space for $k$ Nearest Neighbors | | | | Avg. Euclidean Distance in Pixel Space for $k$ Nearest Neighbors | | | |
|---|---|---|---|---|---|---|---|---|
| | $k=1$ | $k=2$ | $k=5$ | $k=10$ | $k=1$ | $k=2$ | $k=5$ | $k=10$ |
| 3 | 0.8751 | 0.8612 | 0.8567 | 0.8543 | 97.78 | 99.89 | 113.01 | 116.14 |
| 4 | 0.8644 | 0.8541 | 0.8506 | 0.8420 | 106.15 | 106.25 | 113.11 | 115.03 |
| 5 | 0.8428 | 0.8355 | 0.8277 | 0.8265 | 106.62 | 106.87 | 107.25 | 117.01 |
| 6 | 0.8142 | 0.8112 | 0.8056 | 0.8016 | 121.41 | 121.84 | 129.91 | 143.05 |

has almost the same number of parameters as the 5-encoding-block model (see Table A in Appendix) . The results remain largely similar to the small 2-encoding-block model we experimented with earlier which confirms that it is not the number of parameters, but the abstraction level of representations that is responsible for the observed bias amplification. Please note that the results for the classes that are combinations of *genders* and *hair colors* are not reported, because in that case the minority class (blond male) contributes to less that $1\%$ of the training data, and getting a meaningful result of its changes required generating millions of samples, so we reported the results for "gender" classes.

Figure 5 shows 10 consecutive generations of two models with 4 and 6 encoding blocks, trained on CelebA starting from the same seed. In the two highlighted sample pairs, the generated images differ: the deeper model generates examples from majority classes - a "non-blond male" that makes up 40.81% of the dataset, instead of a "blond male", which represents only 0.86% of the dataset; and a "female" that constitutes 58.32% of the dataset, instead of a "male" that accounts for 41.68%. These results are perfectly explained by our bias amplification insight.

### 3.4 "High Variance State" Undermines Privacy Preservation in Diffusion Models

We also investigated the state of "high variance" in which the model generates images that more closely resemble specific samples of the training data. Table 3 gives the average result of measured similarities between the generated images and their nearest neighbors in the training data. For this comparison, cosine similarity in Inception feature space and Euclidean distance in pixel space are used. The experiments are carried on for $k = 1, 2, 5$, and 10. The results clearly show in general, shallower models tend to generate images resembling some specific samples of the training data more than deeper ones, as they yield larger cosine similarities between their features in Inception feature space and their nearest neighbors, while having smaller Euclidean distance in pixel space. This observation emphasizes the vulnerability of shallower models with respect to the privacy of training data in diffusion models.

Also, relevant to privacy preserving of training data is the capability of models in generating diverse images. Figure 4 and Table 2 give average mutual LPIPS scores for generated images by models trained on MNIST and CelebA, respectively. The results consistently show lower LPIPS for shallower models, indicating their limited ability to generate diverse outputs.

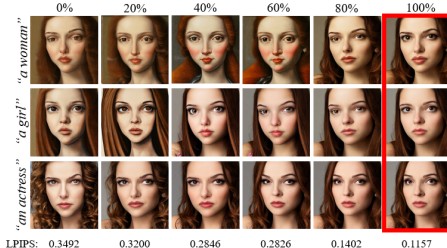

Figure 6: Images generated with SD v1.5 for three similar prompts: "a woman", "a girl", and "an actress", using the same seed (left column). Gradually bypassing the model's mid-block makes the outputs more similar and realistic, as the model relies on less abstract representations, inducing "high-variance" state, where the noisy signal attends to only a few attractors, reflecting specific memorized patterns.

As a qualitative experiment, we also tried to force the pretrained model to work with less abstract data representations to simulate the "high bias" state. To do so, we bypassed the mid-block of the U-Net of SD V1.5. This way, the model relies on the less abstract representations flowing via skip connections. Figure 6 shows the images generated by SD V1.5 for three prompts: "a woman", "a girl", and "an actress". We show the results as percentage bypassed mid-block activations in gradually increasing proportions, up to complete bypassing (100%). As can be observed, the complete bypass results in generating almost similar images for different prompt. This follows naturally from our bias-variance analysis in the previous sections. It entails that promoting higher variance of pre-trained models by bypassing can put the (presumed) privacy of DMs at risk.

## 4 RELATED WORK

**Explanation of Diffusion Models** Many recent studies have focused on explanation of the underlying mechanisms of DMs. Yi *et al.* Yi et al. (2024) showed that the generation of images in DMs starts with low frequency signals in initial steps, followed by the construction of high frequency signals in later steps. This behavior leads to the generation of the overall image first and details later in the denoising process. The memorization-generalization properties of DMs are explored in Li et al. (2023), Li et al. (2024), and Yoon et al. (2023). Furthermore, Vastola Vastola (2025) explored the elements influencing the generalization of DMs. Kwon *et al.* Kwon et al. (2023) uncovered the semantic latent space in DMs, which is further explored in Park et al. (2023) and Wang et al. (2025). As a more intuitive alternative to the common explanation of DM models' generation as *iterative denoising*, Hoover *et al.* Hoover et al. (2023) proposed to conceptualize the inference process as *memory retrieval*.

**Bias Mitigation in Diffusion Models** DMs are accused of amplifying the biases inherent in their training data Friedrich et al. (2023); Bianchi et al. (2023). However, Seshjadri *et al.* Seshadri et al. (2023) challenged this concern and argued that in the text-to-image case, bias exacerbation stems from distribution shift in training captions and prompts. Conversely, as suggested by Chen *et al.*, the bias amplification is more serious than presumed and can potentially results in the cascading of bias towards future modelsChen et al. (2024b).

**Privacy of Diffusion Models** To preserve the confidentiality of potentially private dataset samples, Jahanian *et al.* Jahanian et al. (2022) proposed using generative models to have variants of samples instead of explicitly storing and working with real datasets. However, Carlini *et al.* Carlini et al. (2023) revealed that DMs are prone to generate images highly resembling their training data. Also, it is argued that DMs can be pushed to give away their training samples using membership inference method Duan et al. (2023). In response to the challenge, robust DMs that address privacy concerns are proposed Dockhorn et al. (2023); Wang et al. (2024); Chen et al. (2024a). Moreover, the inherent privacy guarantees and conditions affecting privacy-preserving properties of datasets generated by DMs are analyzed in Wei et al. (2024).

## 5 CONCLUSIONS

In this work, we explored the concept of "bias-variance trade-off" in the context of DMs, which explains bias amplification at one extreme and privacy risks as well as reduced generative diversity at the other. Our arguments are aligned with the opposing concepts of "memorization" and "generalization", but it further captures the state beyond generalization. This study revealed the potential drawbacks of extremely deep DMs, not limited to computational cost. We also showed the vulnerability of pre-trained DMs to be pushed to "high variance" state in order to get information of their training data.

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

# A APPENDIX

## A.1 PROOF OF THEOREM 1

**Theorem 1** (Higher abstraction in representations induces smoother energy landscapes). *Let $E$ : $\mathcal{H} \to \mathbb{R}$ be an energy function defined over a space of representations $\mathcal{H}$, $\phi^{(a)} : \mathcal{H} \to Z^{(a)}$ be a hierarchy of abstraction maps indexed by level $a$, where each $\phi^{(a)}$ is a smooth, contractive transformation, and $Z^{(a)}$ be a vector space of latent representations after some level of abstraction. Define the energy at abstraction level $a$ as*

$$E := E \circ (\phi^{(a)})^{-1} : Z^{(a)} \to \mathbb{R}. \tag{19}$$

*Then the energy landscapes $\{E^a\}_a$ become progressively smoother with increasing $a$ in the following sense:*

$$\left\| \nabla^2 E^{(a+1)} \right\| < \left\| \nabla^2 E^{(a)} \right\|, \quad \text{and} \quad L^{(a+1)} < L^{(a)}, \tag{20}$$

*where $\left\| \nabla^2 E^{(a)} \right\|$ denotes the Hessian norm (curvature), and $L^{(a)}$ is the Lipschitz constant of the gradient $\nabla E^{(a)}$.*

*Proof.* Let $J_{\phi^{(a)-1}}(z^{(a)})$ denote the Jacobian of $\phi^{(a)^{-1}}$ at $z^{(a)}$. Then, by the chain rule, the gradient of the energy function $E^{(a)}$ at $z^{(a)}$ is given by:

$$\nabla E^{(a)}(z^{(a)}) = J_{\phi^{(a)-1}}(z^{(a)})^\top \nabla E(\phi^{(a)^{-1}}(z^{(a)})). \tag{21}$$

Similarly, the Hessian of $E^{(a)}$ is given by:

$$\nabla^2 E^{(a)}(z^{(a)}) = J_{\phi^{(a)-1}}(z^{(a)})^\top \nabla^2 E(\phi^{(a)^{-1}}(z^{(a)})) J_{\phi^{(a)-1}} + \sum_i \frac{\partial E}{\partial x_i} \nabla^2 \phi^{(a)^{-1}}(z^{(a)}). \tag{22}$$

Taking norms on both sides, we obtain:

$$\left\| \nabla^2 E^{(a)}(z^{(a)}) \right\| \leq \overbrace{\| J_{\phi^{(a)-1}} \|^2}^{\text{decreasing in } a} \cdot \overbrace{\left\| \nabla^2 E \right\|}^{\text{fixed}} + \| \sum_i \overbrace{\frac{\partial E}{\partial x_i}}^{\text{fixed}} \overbrace{\nabla^2 \phi^{(a)^{-1}}(z^{(a)})}^{\text{decreasing in } a} \|. \tag{23}$$

Based on Lemma 1, $\phi^{(a)}$ is a contractive and smoothing map (i.e., the Jacobian norm decreases and the second derivatives of its inverse diminish with increasing $a$), both terms in the bound decrease with increasing $a$. Therefore, the norm of the Hessian satisfies

$$\left\| \nabla^2 E^{(a+1)} \right\| < \left\| \nabla^2 E^{(a)} \right\|. \tag{24}$$

Moreover, let $L^{(a)}$ denote the Lipschitz constant of $\nabla E^{(a)}$, so that:

$$\left\| \nabla E^{(a)}(z_1^{(a)}) - \nabla E^{(a)}(z_2^{(a)}) \right\| \leq L^{(a)} \left\| z_1^{(a)} - z_2^{(a)} \right\|, \quad \forall z_1^{(a)}, z_2^{(a)} \in Z^{(a)}. \tag{25}$$

Since $\left\| \nabla^2 E^{(a)} \right\| \geq L^{(a)}$, Equation 25 implies:

$$L^{(a+1)} < L^{(a)} \tag{26}$$

This shows that both the second-order and first-order variations of the energy function diminish with increasing abstraction level. Hence, the energy landscapes become progressively smoother. $\square$

**Lemma 1** (Higher levels of abstraction in the representation induce smaller Jacobian norm and second-order derivative of the representation mapping).

*Proof.* **Diminishing Jacobian:** Define the sensitivity to perturbations at level $a$ as the norm of the Jacobian as:

$$J(a) := \left\| \frac{\partial z^{(a)}}{\partial x} \right\|, \quad J(a+1) := \left\| \frac{\partial z^{(a+1)}}{\partial x} \right\|. \tag{27}$$

By first-order Taylor expansion, the effect of the perturbation on the representation at level $a$ is:

$$z^{(a)}(x + \delta) \approx z^{(a)}(x) + \frac{\partial z^{(a)}}{\partial x}\delta, \tag{28}$$

and thus the magnitude of the change is:

$$\left\|z^{(a)}(x + \delta) - z^{(a)}(x)\right\| \approx \left\|\frac{\partial z^{(a)}}{\partial x}\delta\right\| \leq J(a).\|\delta\|. \tag{29}$$

Similarly, for level $a + 1$:

$$z^{(a+1)}(x + \delta) \approx z^{(a+1)}(x) + \frac{\partial z^{(a+1)}}{\partial x}\delta, \tag{30}$$

and

$$\left\|z^{(a+1)}(x + \delta) - z^{(a+1)}(x)\right\| \approx \left\|\frac{\partial z^{(a+1)}}{\partial x}\delta\right\| \leq J(a+1).\|\delta\|. \tag{31}$$

As increasing abstraction decreases sensitivity to perturbations:

$$\left\|z^{(a+1)}(x + \delta) - z^{(a+1)}(x)\right\| < \left\|z^{(a)}(x + \delta) - z^{(a)}(x)\right\|. \tag{32}$$

Using the approximations in Equation (29) and Equation (31), this gives:

$$J(a+1).\|\delta\| < J(a).\|\delta\| \implies J(a+1) < J(a). \tag{33}$$

**Diminishing second derivative of $\phi^{(a)^{-1}}$:** Consider a small perturbation $\delta$ added at level $a + 1$:

$$z^{(a+1)} + \delta. \tag{34}$$

The corresponding change at level $a$ is approximated by a first and second order Taylor expansion of the inverse map $\phi^{(a+1)^{-1}}$:

$$z^{(a)} + \Delta z^{(a)} \approx \phi^{(a+1)^{-1}}(z^{(a+1)} + \delta) = z^{(a)} + J_{\phi^{(a+1)^{-1}}}\delta + \frac{1}{2}\delta^\top \nabla^2 \phi^{(a+1)^{-1}}\delta, \tag{35}$$

where $J_{\phi^{(a+1)^{-1}}}$ is the Jacobian matrix and $\nabla^2\phi^{(a+1)^{-1}}$ is the Hessian tensor of $\phi^{(a+1)^{-1}}$. The change in the energy at level $a + 1$ due to $\delta$ is:

$$\Delta^2 E^{(a+1)} \approx \delta^\top \nabla^2 E^{(a+1)}\delta. \tag{36}$$

At level $a$, the change is:

$$\Delta^2 E^{(a)} \approx \left(J_{\phi^{(a+1)^{-1}}}\delta\right)^\top \nabla^2 E^{(a)}\left(J_{\phi^{(a+1)^{-1}}}\delta\right), \tag{37}$$

ignoring higher order terms involving $\nabla^2\phi^{(a+1)^{-1}}$. As increasing abstraction implies reduction in sensitivity, we have $|J_{\phi^{(a+1)^{-1}}}| < 1$. So,

$$\|\nabla^2 E^{(a+1)}\| \leq \|J_{\phi^{(a+1)^{-1}}}\|^2.\|\nabla^2 E^{(a)}\|. \tag{38}$$

Thus,

$$\|\nabla^2 E^{(a+1)}\| < \|\nabla^2 E^{(a)}\| \tag{39}$$

$\square$

## A.2 Implementation Details

All the experiments are implemented in Pytorch and they used three NVIDIA RTX 3090 GPUs each with 24G RAM. The details of the trained models are summarized in Table A.

## A.3 Bias Amplification

Figure A and Figure B show qualitative examples of bias amplification in deeper models. Figure A shows representative examples where shallower models generated images of "blond-male" minority group, but the images got changed to the majority groups (classes) of "non-blond male" and "blond female" for the deeper models when starting from the same seeds. Similarly, Figure B presents cases in which shallow models generate "male" images that are minority group, while deeper models converted the same seeds to the majority group, i.e., "female".

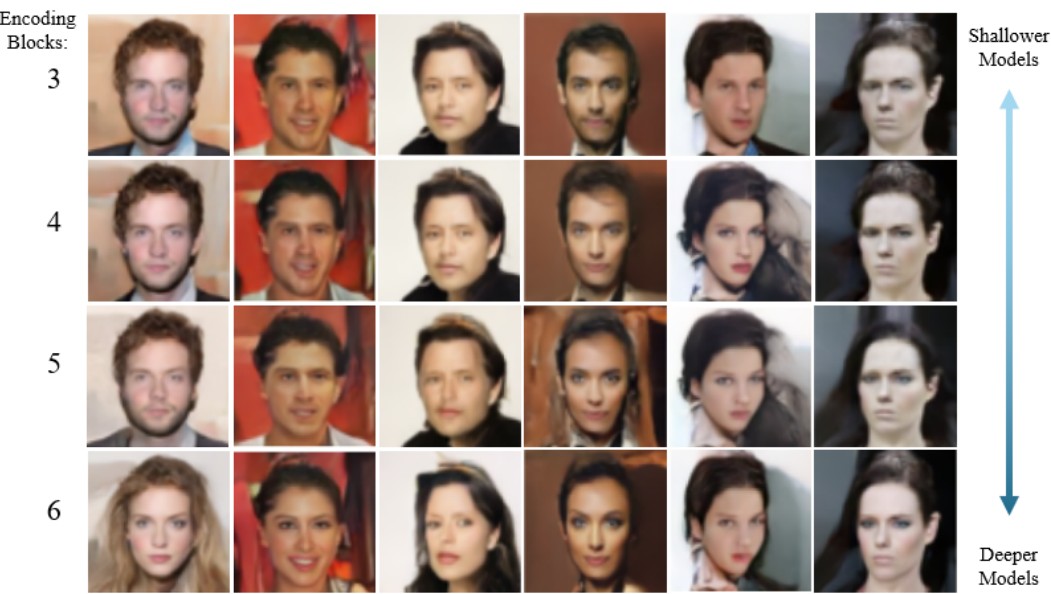

Figure A: Bias amplification in deeper models. Top row shows samples of generated images resembling "blond male" by a relatively shallow model consisting of three encoding blocks. Lower rows show generated images of increasingly deeper models starting from the same seeds. It can be seen that the deeper models prefer generating samples from the majority groups (non-blond male and blond-female) instead of the minority group (blond-male), thereby amplifying the data bias.

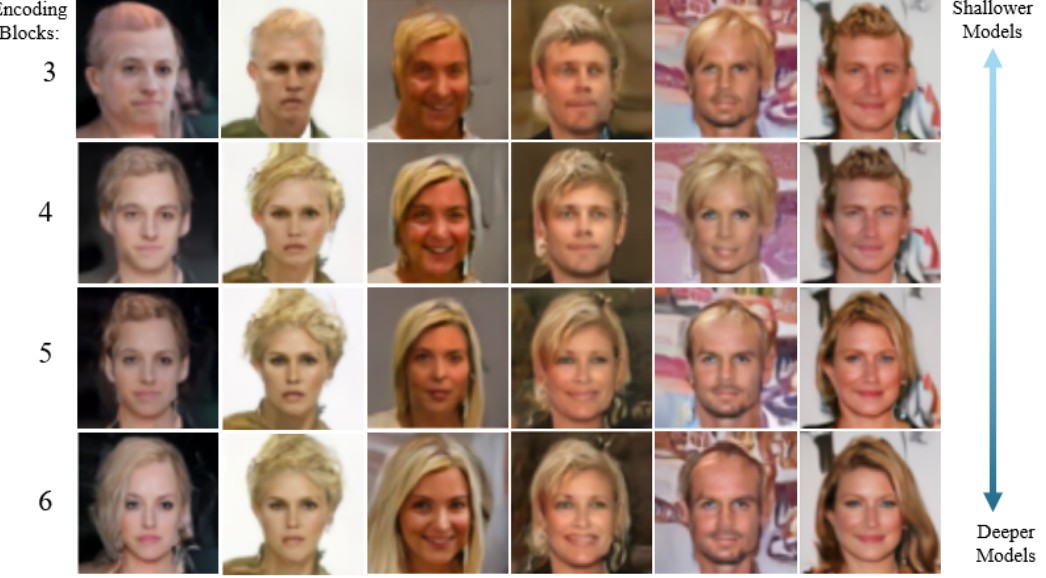

Figure B: Representative cases where shallow models generate samples of minority group "male", but deeper models generate majority group "female" starting from the same seeds.

Table A: Details of trained diffusion models. Please note the number of parameters on the "large 2 encoding blocks" model, which is at the same scale of deeper models ("5 and 6 encoding blocks"). However, its bias-amplification behavior is more similar to shallower models ("2 and 3 encoding blocks") (see Table 2 in the main paper).

| Number of encoding blocks | Architecture | | | | |
|---|---|---|---|---|---|
| | Down-block types | Up-block types | Block out channels | Layers per block | # of parameters |
| 2 | DownBlock2D, AttnDownBlock2D | AttnUpBlock2D, UpBlock2D | 32, 64 | 2 | 1002883 |
| 2 (Large) | DownBlock2D, AttnDownBlock2D | AttnUpBlock2D, UpBlock2D | 128, 256 | 2 | 15930883 |
| 3 | DownBlock2D, AttnDownBlock2D, DownBlock2D | UpBlock2D, AttnUpBlock2D, UpBlock2D | 32, 64, 128 | 2 | 3692867 |
| 4 | DownBlock2D, DownBlock2D, AttnDownBlock2D, DownBlock2D | UpBlock2D, UpBlock2D, AttnUpBlock2D, UpBlock2D | 32, 32, 64, 128 | 2 | 3859203 |
| 5 | DownBlock2D, DownBlock2D, DownBlock2D, AttnDownBlock2D, DownBlock2D | UpBlock2D, AttnUpBlock2D, UpBlock2D, UpBlock2D, UpBlock2D | 32, 64, 128, 128, 256 | 2 | 16887619 |
| 6 | DownBlock2D, DownBlock2D, DownBlock2D, DownBlock2D, AttnDownBlock2D, DownBlock2D | UpBlock2D, AttnUpBlock2D, UpBlock2D, UpBlock2D, UpBlock2D, UpBlock2D | 32, 64, 128, 128, 256, 256 | 2 | 27290691 |

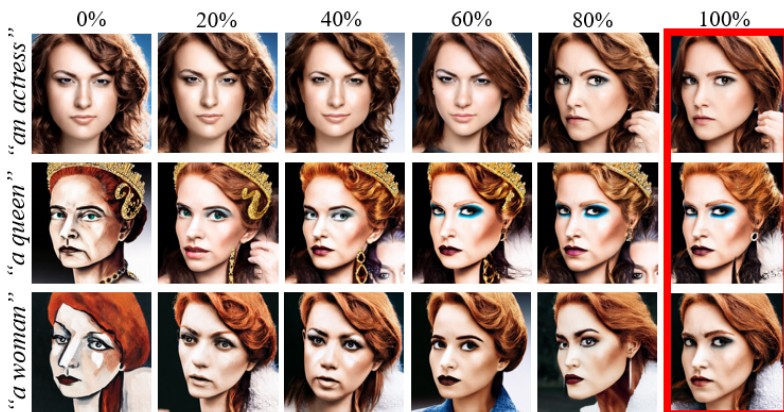

Figure C: A model pushed towards "high variance" state by bypassing its mid-block generates similar looking images for slightly different prompts. The resulting images not only have high perceptual similarity across the prompts, but also better realism. This behavior is consistent with our arguments about "high variance" state, in which the model attends to fewer memories.

## A.4 PUSHING MODELS TO HIGH VARIANCE STATE

Illustrated in Figure C, the model generates slightly different images for prompts "an actress", "a queen", , and "a woman". Then, by gradually bypassing the mid-block, the model gets pushed to "high variance" state. Correspondingly, the generated images begin to look more alike and realistic in the "high variance" state, supporting our claim of attending to lower number of memories in this state.

