# OpenReview forum: "Intriguing Bias-Variance Tradeoff in Diffusion Models"
_ICLR.cc/2026/Conference — Submitted to ICLR 2026_

### Official Review · Reviewer_Gxyj · 2025-10-25

**Soundness:** 3
**Presentation:** 4
**Contribution:** 3
**Rating:** 8
**Confidence:** 4

**Summary:**

The paper studies the “bias–variance tradeoff” in diffusion models, inspired by the k-NN analogy, showing that the abstraction level of latent representations controls the smoothness of the learned energy landscape.
Models operating at higher abstraction levels yield smoother landscapes that amplify dataset bias, whereas models relying on lower-level, less abstract features tend to memorize training samples and risk privacy leakage.
Both theoretical proofs and experiments on MNIST, CelebA, and Stable Diffusion v1.5 empirically support this connection.

**Strengths:**

It offers a novel and unified theoretical framework that connects bias amplification and memorization in diffusion models through the bias–variance tradeoff.

**Weaknesses:**

1. The experimental validation is somewhat limited — results are based on small-scale datasets and qualitative observations on Stable Diffusion v1.5,  and only train on the unconditional model. If you
2. The definition of the “abstraction level” is not clearly established. The paper implicitly equates abstraction with the number of encoder blocks or latent downsampling, but this connection is heuristic rather than theoretically grounded

**Questions:**

1. Have you examined whether similar bias–variance phenomena appear in Rectified Flow models ?
2. In DiT-style architectures without a clear encoder–decoder hierarchy, how can the notion of “abstraction level” be rigorously defined or measured?

---

> ### Author Response · Authors · 2025-11-23
> **Response to Reviewer Gxyj**
>
> We thank the reviewer for their positive assessment of our work and **recognizing the novelty and contribution** of the proposed theoretical framework. The responses to the reviewer's comments are provided as follows.
>
> **Weaknesses:**
>
> **W1. The experimental validation is somewhat limited — results are based on small-scale datasets and qualitative observations on Stable Diffusion v1.5, and only train on the unconditional model.**
>
> **Answer:** We appreciate the reviewer’s perspective. In response, we designed further experiments on a larger scale dataset (Stanford Dogs Dataset). The experiments are being conducted and we will report the results shortly.
>
> **W2. The definition of the “abstraction level” is not clearly established. The paper implicitly equates abstraction with the number of encoder blocks or latent downsampling, but this connection is heuristic rather than theoretically grounded**
>
> **Answer:** We thank the reviewer for raising this point. We will add explicit definition of abstraction as the result of aggregation, averaging, pooling, normalization, or attention-weighted summarization of representations, as they remove local detail and retain only global and collective information. This is exactly the mechanism behind abstraction in both CNN-based and Transformer-based diffusion models.
>
> We will add this definition to the paper:
>
> **Definition:** (Abstraction Levels of Representations)
>
> Let $Z$ be a space of an intermediate model representations, and let $f: Z \to Z'$ be a mapping that transforms representations $z \in Z$ into new representations $z' = f(z) \in Z'$.
>
> We define representations in $Z'$ are of a higher **level of abstraction** than those in $Z$ if $f$ aggregates multiple low-level details into fewer, more meaningful features, producing a coarse-grained, semantically enriched view of the input.
>
> **Questions:**
>
> **Q1. Have you examined whether similar bias–variance phenomena appear in Rectified Flow models?**
>
> **Answer:** Great point. We appreciate this comment and view it as a valuable direction for further exploration. Based on our current understanding of the bias–variance tradeoff discussed in the paper, we anticipate that similar phenomena should also arise in Rectified Flow (RF) models. Although RF models primarily differ in the shape of their generative trajectories (straight rather than curved), the progressive abstraction of representations remains largely unchanged. Since our formulation of the bias–variance tradeoff is driven by the level of abstraction across these representations—rather than the specific trajectory parameterization—we expect RF models to exhibit the same underlying behavior.
>
> **Q2. In DiT-style architectures without a clear encoder–decoder hierarchy, how can the notion of “abstraction level” be rigorously defined or measured?**
>
> **Answer:** We thank the reviewer for raising this insightful question. In DiT-style architectures, the absence of an explicit encoder–decoder hierarchy does not eliminate the notion of abstraction. Transformer layers inherently induce a progression of abstraction through their global receptive field and iterative mixing of information. Self-attention enables each token to integrate information from the entire spatial domain—often even more comprehensively than the strictly local receptive fields of convolutional layers in U-Nets [1]. As depth increases, tokens aggregate information from progressively larger and more diverse context sets, producing higher-level, more invariant, and more semantically enriched representations. This structured evolution of representations therefore serves the same functional role as increasing abstraction in hierarchical encoder–decoder architectures.
>
> **References:**
>
> [1] M. Raghu et al., Do vision transformers see like convolutional neural networks? NeurIPS, 2021.

---

> ### Author Response · Authors · 2025-11-26
> **Additional Experiment**
>
> In response to Weakness 1, as an additional experiments, we fine-tuned 3 different variations of SD V1.5 on a biased custom dog breed dataset with images from ImageNet. The dataset is customized as follows:
>
> * We took Stanford Dogs with 120 classes of dog breeds taken from ImageNet.
> * Then we introduced bias in the dataset by keeping \(n_c + 20\) samples of each class, where \(n_c\) is the class number. The classes are ordered alphabetically. So, there are 21 up to 140 samples of each class according to their class number.
> * The models are finetuned each for 100 epochs and the learning rate of 1e-8.
>
> The fine-tuned models are:
> * $M_1$: The original model
> * $M_2$: The original model with its U-Net bottleneck layer pruned (less abstract representation)
> * $M_3$: The original model with the same number of bottleneck parameters pruned (97038080), 11.29\% of total model parameters, randomly from the entire U-Net
>
> |                 |               |        |        |        |        |    Class     | Number |        |         |       |        |        |        |
> |:---------------:|:-------------:|:------:|:------:|:------:|:------:|:------:|:------------:|:------:|:------:|:------:|:------:|:------:|:------:|
> |                 | \# of params. |   10   |   20   |   30   |   40   |   50   |      60      |   70   |   80   |   90   |   100  |   110  |   120  |
> |  Complete Model |   859520964   | -4.8\% | -3.1\% | -2.0\% | -2.1\% | -2.3\% |    -2.6\%    | +3.3\% | +3.5\% | +4.6\% | +5.1\% | +5.5\% | +6.7\% |
> |   No Midblock   |   762482884   | -3.9\% | -4.0\% | -1.7\% | -1.3\% | -0.6\% |    +0.1\%    | +4.0\% | +3.4\% | +4.1\% | +4.9\% | +5.4\% | +6.1\% |
> | Randomly Pruned |   762482884   | -4.5\% | -3.7\% | -2.2\% | -2.0\% | -1.5\% |    -1.1\%    | +2.6\% | +3.7\% | +4.6\% | +5.0\% | +5.9\% | +6.4\% |
>
> Using each model, 20000 images were generated with the prompt "a dog". The generated images are then classified using a pretrained dog breed classifier taken from "https://huggingface.co/jhoppanne/Dogs-Breed-Image-Classification-V1". We report, for 10 representative classes, the change in the number of generated images corresponding to each class across the finetuned models. As it can be seen, the model without mid-block layer (less depth and less abstraction) became less biased compared the other two. Furthermore, the randomly pruned model that has the exact same number of parameters the model with no mid-block has almost the same behavior in introducing bias as the complete model with the same depth.

---

> > ### Comment · Reviewer_Gxyj · 2025-11-27
> >
> > The authors’ response has addressed my earlier concerns. I will keep my original score of 8

---

> > > ### Author Response · Authors · 2025-11-27
> > >
> > > Thank you very much. We sincerely appreciate your acknowledgment of the strengths of our work and the recommendation for acceptance.

---

### Official Review · Reviewer_rj1q · 2025-10-30

**Soundness:** 2
**Presentation:** 2
**Contribution:** 2
**Rating:** 4
**Confidence:** 3

**Summary:**

This work uncovers a general bias-variance trade-off phenomenon, based on the landscape analysis of hierarchy representations. The basic  insight is: (i) deep models would emphasize majority sample classes and generate more diverse images; (ii) shallow models are more sensitive and hence show privacy risks.

**Strengths:**

1. The paper is clearly organized with many illustrations.
2. Both theoretical analysis and multiple numerical verifications are provided to support main findings.
3. Main findings are new and interesting.

**Weaknesses:**

1. It seems that the whole framework (e.g. Eq. (3, 4, 5)), analysis and results (e.g. Thm. 1) work for general NNs/en-decoders. Why do authors only discuss diffusion models, and are there any new insights specifically tailored to diffusion models?
2. Following 1, it would be better to provide self-contained formulations of e.g. latent spaces and energy landscapes, particularly for diffusion models, and formulate the exact definition of "abstraction" repeatedly used in this paper.
3. For Thm. 1:
- The notations $E^a$ or $E^{(a)}$ should be consistent, and in fact, authors do not define $E^{(a)}$ in Eq. (6).
- The transformation $\phi^{(a)}$:
    - $\phi^{(a)}$ is assumed as a contractive mapping, but why does it hold particularly for diffusion models? It seems that this is due to the statement "As increasing abstraction decreases sensitivity to perturbations" in the proof, but why does this hold?
    - $\phi^{(a)}$ is also assumed as a invertible mapping in Eq. (6). Why does it hold particularly for diffusion models?
4. This work aims to justify both bias and variance in diffusion models, and the bias aspect is mathematically justified in Thm. 2. However, this is not the case for the variance aspect. Can authors give more quantitative analysis of these high-variance cases? In addition, it is not clear how these theoretical/numerical justifications of bias and variance relate to their definitions Eq. (4, 5).
5. As shown in e.g. Fig. 4 and Tab. 3, deep models appear both high biases and diversity. However, biased models prefer majority classes, which reduce the overall diversity in principle. Can authors clearly explain the definition of diversity used and evaluated in this work, otherwise the implications would be contradictory.
6. As discussed in this work, deep models show high biases on majority classes, and shallow ones have high variance and hence privacy risks. Then, as required by the paper title, how can we trade-off the depth in practice, particularly for diffusion models?

**Questions:**

See Weaknesses.

---

> ### Author Response · Authors · 2025-11-23
> **Response to Reviewer rj1q (1/3)**
>
> We appreciate the reviewer’s comments and thank them for highlighting **the clarity of the presentation**, **the strength of the theoretical and empirical analysis**, and the **novelty of our findings**. We address the reviewer’s concerns and provide our detailed responses below.
>
> **Weaknesses:**
>
> **W1. It seems that the whole framework (e.g. Eq. (3, 4, 5)), analysis and results (e.g. Thm. 1) work for general NNs/en-decoders. Why do authors only discuss diffusion models, and are there any new insights specifically tailored to diffusion models?**
>
> **Answer:** We thank the reviewer for this insightful comment. You are right, our theoretical framing of the bias-variance phenomenon is broader than just diffusion models. However, in this paper we specify it to follow hierarchical denoising based neural architecture in generative domain for diffusion models only, and later provide empirical evidence focusing on diffusion models only. We focus on diffusion models because these are currently considered state-of-the-art models in image generation, while the theoretical understanding of their generative behavior is limited. Their wide applicability in different domains is causing considerable bias and privacy issues, among others. Hence, theoretical treatment of their generative behavior is currently critical, which is the intended contribution of this work. While it is not possible to cover all NNs/en-decoders in one paper, we do plan to extend our theory to other model types in the future.
>
> **W2. Following 1, it would be better to provide self-contained formulations of e.g. latent spaces and energy landscapes, particularly for diffusion models, and formulate the exact definition of "abstraction" repeatedly used in this paper.**
>
> **Answer:**  We thank the reviewer for this constructive suggestion. To address it, we provide self-contained definitions of abstraction levels and energy landscapes corresponding to different abstraction levels of  representations in diffusion models. We will add these definitions to the paper.
>
> **Definition:** (Abstraction Levels of Representations)
>
> Let $Z$ be a space of an intermediate model representations, and let $f: Z \to Z'$ be a mapping that transforms representations $z \in Z$ into new representations $z' = f(z) \in Z'$.
>
> We define representations in $Z'$ are of a higher **level of abstraction** than those in $Z$ if $f$ aggregates multiple low-level details into fewer, more meaningful features, producing a coarse-grained, semantically enriched view of the input.
>
> **Definition:** (Energy Landscapes Corresponding to Abstraction Levels of Representations in Diffusion Models)
>
> Let $Z^{(a)}$ denote the space of representations at abstraction level $a$ of a diffusion model.
> An **energy landscape** $E^{(a)}$ at abstraction level $a$ is a function that assigns a real-valued energy to each representation $z \in Z^{(a)}$: $E^{(a)} : Z^{(a)} \to \mathbb{R}.$
>
> **W3. For Thm. 1:**
>
> * **The notations \(E^a\) or \(E^{(a)}\) should be consistent, and in fact, authors do not define $E^{(a)}$ in Eq. (6).**
>
> **Answer:** Thank you for pointing this out. We will fix it.
>
> * **The transformation $\phi^{(a)}$**
>
>     + **$\phi^{(a)}$ is assumed as a contractive mapping, but why does it hold particularly for diffusion models? It seems that this is due to the statement "As increasing abstraction decreases sensitivity to perturbations" in the proof, but why does this hold?**
>
>     **Answer:** In diffusion models, the encoder progressively abstracts features from the input by aggregating information across spatial locations and channels through downsampling, attention, and normalization operations. These mechanisms reduce sensitivity to fine-grained variations and noise, ensuring that small differences in the input produce proportionally smaller differences in the latent representation. Consequently, the abstraction maps $\phi^{(a)}$ are contractive along low-level detail directions, which justifies the assumption in the context of diffusion models.

---

> ### Author Response · Authors · 2025-11-23
> **Response to Reviewer rj1q (2/3)**
>
> **W3. continued**
>
> * **The transformation $\phi^{(a)}$:**
>
>     + **$\phi^{(a)}$ is also assumed as a invertible mapping in Eq. (6). Why does it hold particularly for diffusion models?**
>
> **Answer:** In our framework, we consider a hierarchy of energy landscapes that become increasingly abstract at deeper levels. Correspondingly, we define a hierarchy of abstraction mappings that transform a less abstract landscape into a more abstract one, with the formal inverses performing the opposite transformation. While the exact forms of these abstraction mappings are unknown, the model effectively learns to approximate them during training.
>
> In an encoding-decoding paradigm, a well-trained model is expected to effectively map inputs to latent representations and back. This justifies the assumption of invertibility: although we do not require explicit knowledge of the inverse mapping or perfect reconstruction, the invertibility assumption provides a principled way to compare representations across abstraction levels. In other words, it serves as a conceptual tool for analysis rather than a strict requirement on the model’s implementation.
>
> **W4. This work aims to justify both bias and variance in diffusion models, and the bias aspect is mathematically justified in Thm. 2. However, this is not the case for the variance aspect. Can authors give more quantitative analysis of these high-variance cases? In addition, it is not clear how these theoretical/numerical justifications of bias and variance relate to their definitions Eq. (4, 5).**
>
> **Answer:** Theorem 1 establishes that higher levels of abstraction lead to smoother energy landscape, which implies that in low abstraction levels the initial sampled noise will converge to a minimum corresponding to a small number of implicit memories, and conversely in high levels of abstraction the initial sampled noise will converge to a minimum corresponding to a larger number of memories. It is already justifies the high variance state, as in low abstraction levels the output is sensitive to the small number of memories (high variance). However, to justify the high bias state we had to take a further step and show that the "merged local minima" are biased representatives and lead to amplification of existing bias (otherwise there would be no high bias).
>
> Table 3 in the paper also gives quantitative results related to high variance state. It compares the similarity of generated images both in pixel space and feature space with their nearest samples in the training dataset. The shallower models exhibit more similarity to the training data samples, which we interpreted as high variance state.
>
> Eq. 4 and Eq. 5 are fundamental formulations of bias and variance, respectively \cite{geman1992neural}. Eq. 4 defines bias as the distance between the expectation of generated image and training data distribution, which is exactly what we measured and compared in Table 2 and Fig. 4.
>
> Eq. 4:
> $$ \mathrm{bias}(\hat{f}(x_0)) = \overbrace{\mathbb{E}[\hat{f}(x_0)]}^{\text{expectation of model output}} - \overbrace{f(x_0)}^{\text{true underlying distribution}},$$
>
> Eq. 5 defines variance as the expected deviation of specific model outputs from its expected output. This captures how sensitive the model is to variations in the training data. To connect this to diffusion models, we quantify variance empirically by measuring how similar generated images are to the training samples, reported in Table 3.
>
> Eq. 5:
> $$ \mathrm{variance}(\hat{f}(x_0)) = \overbrace{\mathbb{E}[(\overbrace{\hat{f}(x_0)}^{\text{specific output}}- \overbrace{\mathbb{E}[\hat{f}(x_0)]}^{\text{expected output}})^2}^{\text{expected squared deviation}}].$$
>
> Please note that, to emphasize the connection between the *k*-NN perspective and our work, we use “classes” and “concepts” interchangeably (see the footnote on page 3 of the paper).
>
> **W5. As shown in e.g. Fig. 4 and Tab. 3, deep models appear both high biases and diversity. However, biased models prefer majority classes, which reduce the overall diversity in principle. Can authors clearly explain the definition of diversity used and evaluated in this work, otherwise the implications would be contradictory.**
>
> **Answer:** Thank you for raising this point. We used  average LPIPS metric measured between generated image pairs, which show the diversity in feature space. We will add this sentence to paper explicitly: "We refer to diversity in feature space not in label space.''

---

> ### Author Response · Authors · 2025-11-23
> **Response to Reviewer rj1q (3/3)**
>
> **W6. As discussed in this work, deep models show high biases on majority classes, and shallow ones have high variance and hence privacy risks. Then, as required by the paper title, how can we trade-off the depth in practice, particularly for diffusion models?**
>
> **Answer:**  We thank the reviewer for this important question. This paper aims to highlight the trade-off between high bias and high variance in diffusion models and its relation to the abstraction levels of representations. Considering this, developers should be aware and take control of the trade-off based on their priorities in their specific applications. They can decide whether privacy is more of concern or mitigating bias.
>
> **References:**
>
> [1] S. Geman et al., Doursat. Neural networks and the bias/variance dilemma. Neural computation, 1992.

---

### Official Review · Reviewer_KS7X · 2025-10-31

**Soundness:** 2
**Presentation:** 3
**Contribution:** 2
**Rating:** 2
**Confidence:** 4

**Summary:**

This paper studies the bias-variance tradeoff in diffusion models. They first theoretically demonstrate that a energy model with higher abstraction level tends to have more bias, i.e., the majority class will be amplified. On the other hand, low abstraction leads to local minimums close to the individual data, hence suffer from high variance. They validate these conceptions in diffusion models. They find that diffusion models with more encoding blocks (higher abstraction level) exhibits bias, generating more from majority class, whereas models with less encoding blocks (lower abstraction level) exhibits high variance, generating samples that are more similar to the training data.

**Strengths:**

The empirical observations on the bias-variance tradeoff in diffusion models are interesting, which can inspire further investigation.

**Weaknesses:**

1. The energy model and KNN perspective (figure 2, table 1) of diffusion models are mostly conceptual, with insufficient empirical or theoretical support. Specifically, the theory has nothing to do with diffusion models, while I understand the conceptual similarity, it still feels kind of vague. Overall, I feel this work is not rigorous enough, and not very convincing.

2. The experiment design could be better. For example, it is unclear whether when increasing the encoder block, the total number of model parameters is fixed or not. It is well-known that larger models will be more prone to memorization, i.e., higher variance. How to isolate the effect of abstraction (number of encodings blocks) from the effects of total model parameters? I understand one related experiment is provided, but the setting in others remain unclear.

3. The experiment is limited to simple dataset such as MNIST and face dataset. The number of classes are limited. To support the arguments in the paper, more experiments should be performed on dataset with large number of classes such as ImageNet.

**Questions:**

1. Why more encoding blocks leads to higher abstraction level? Imagine given enough model parameters, one encoding block can learn the same function as multiple encoding blocks.

2. Why more encoding blocks lead to less variance, which increase the model parameters? This seems to contradict with prior works that demonstrate a larger model tends to memorize more (have higher variance).

3. No matter how many encodings blocks a model have, when trained with denoising score matching loss, they are always approximating the data distribution, which in theory should have no bias and variance issue. It is unclear why the observed bias-variance tradeoff emerges.

4. Can you do more experiments on dataset with more classes?

---

> ### Author Response · Authors · 2025-11-23
> **Response to Reviewer KS7X (1/3)**
>
> We thank the reviewer for their feedback and **finding our work interesting**, and **acknowledging that it can inspire further investigations**. We respond to each comment below.
>
> **Weeknesses:**
>
> **W1. The energy model and KNN perspective (figure 2, table 1) of diffusion models are mostly conceptual, with insufficient empirical or theoretical support. Specifically, the theory has nothing to do with diffusion models, while I understand the conceptual similarity, it still feels kind of vague. Overall, I feel this work is not rigorous enough, and not very convincing.**
>
> **Answer:**
> We appreciate the reviewer's viewpoint. The energy landscape and *k*-NN perspectives presented in Figure 2 and Table 1 are intended to provide an intuitive grounding for our core theoretical and empirical analysis that leads to interesting observations - as kindly acknowledged by the reviewer. While these perspectives are conceptual, they are well-supported by existing literature and align with commonly accepted observations.
>
> We intentionally avoided introducing overly complex mathematics in this part to maintain the focus on the core theoretical insights and empirical findings.
>
> In response to the reviewer’s comment, we will enhance Table 1 (see below) to include references supporting each perspective we discuss. Similarly, we will list the relevant references in Figure 2 to clarify the sources and connections. We hope these updates improve clarity and address the concerns regarding rigor and support.
>
> In the following we mentioned the updates in Table 1 and references and additional explanation to clarify Figure 2.
>
> * **Table 1:**
>
> |                            | *k*-NN *[3]*                                                                          | DM *[4]*                                                                                                                                   |
> |----------------------------|----------------------------------------------------------------------------------------------------------------------------------|------------------------------------------------------------------------------------------------------------------------------------------------------------------------------|
> | Storing training data      | explicitly stores the training data *[3, 1]*             | implicitly stores the training data *[2, 6]*                                                                          |
> | Inference initialized with | a query *[3, 1]*                                         | a corrupted signal *[4, 11]*                                                                             |
> | Inference process          | attends to nearest neighbors *[3, 1]*                    | converges to a minimum corresponding to nearby memories *[6]*                                                                             |
> | Smoothing factor           | *k* directly defined as a hyperparameter *[3, 1]* | $\tau$ defined by abstraction level of representation *[ 6, 5]*                                                     |
> | Larger smoothing factor    | smoothens the decision boundary *[3, 1]*                 | smoothens the energy landscape *Our hypothesis - Shown by Theorem 1*                                                                                        |
> | Inference goal             | classify the query *[3, 1]*                              | refine the corrupted signal to an image resembling [a class in] training data *[4, 11]* |
>
> * **Fig.2 Part A)**
>     + Associative Memory [9, 7, 10, 8]
>     + *k*-Nearest Neighbors [1]
>     + Diffusion Models [6, 2]
> * **Fig. 2 Part B) (High Bias Vs. High Variance)**
>     + High Bias (green background): Representatives of experimental result on High Bias and illustration of high bias state (can be compared to *k*-NN). Theorem 1 and Theorem 2 support theoretically, and supporting results are given in Table 2 and Figure 5.
>     + High Variance (purple background): Illustration of high variance state. The theoretical discussion in Theorem 1 and Theorem 2, and experimental results given in Table 3 and Figure 6 support it.
> * **Fig. 2 Part C)** An illustration showing that our definition of High Variance in DMs corresponds to Memorization which perfectly aligns with the literature and the definition of High Bias extends the current known range of memorization-generalization. Up to this point it is a hypothesis which is later shown to hold throughout the paper.
>
> The theory provided in the paper is specifically related to DMs. The relation of Eq. 1 and Eq. 2, which show core definition of *k*-NN might seem unclear to diffusion models at first glance, however, these are provided to show the connection between *k* in *k*-NNs and $tau$ in energy models. Eq. 4 and Eq. 5, are equations defining bias-variance dilemma, which are not restricted to *k*-NNs.

---

> ### Author Response · Authors · 2025-11-23
> **Response to Reviewer KS7X (2/3)**
>
> **W2. The experiment design could be better. For example, it is unclear whether when increasing the encoder block, the total number of model parameters is fixed or not. It is well-known that larger models will be more prone to memorization, i.e., higher variance. How to isolate the effect of abstraction (number of encodings blocks) from the effects of total model parameters? I understand one related experiment is provided, but the setting in others remain unclear.**
>
> **Answer:** Thank you very much for raising this interesting point. Actually, our experiments carefully account for the phenomenon that 'larger models may be prone to memorization'. For all our experiments, we ensure that the models are well trained and they are not under- or over-parameterized. Throughout our results, it can be verified that the models maintain the overall performance and image quality. Please note, we have also reported the number of parameters of the models in Table A (appendix). It is clear from the Table that the model parameters increase with encoder blocks, however not drastically. This is because we base our analysis on well-trained  models only, that retain the original performance. We also stress that the case of 2(Large)-encoding blocks already covers the model size up to 5 blocks - see Table A (appendix). A corresponding 3(Large) size would cause over-parameterization problem, besides exceeding our analysis limit. Hence, we provide results for 2(Large) only. This is not an insufficiency in our experiments, rather an indication of our careful analysis. Given that all models are well-trained and they retain similar overall generative performance, the impact of variations in the output distributions are naturally isolated to the effects of abstraction.
>
> **W3. The experiment is limited to simple dataset such as MNIST and face dataset. The number of classes are limited. To support the arguments in the paper, more experiments should be performed on dataset with large number of classes such as ImageNet.**
>
> **Answer:** We thank the reviewer for the comment. While we are currently running experiments to address this comment, we would like to clarify that our choice of datasets was based on common practices in the literature. MNIST is one of the most widely used dataset for foundational analysis like ours, and human vision is known to be highly  sensitive to facial features. Hence, both datasets provide ideal grounding to fully expose the studied bias-variance relation with a clear analysis. Whereas larger datasets might be trendy in other domains, their benefits in foundational property analysis is limited - to say the least. Not to mention, our analysis requires training multiple diffusion models, which makes the use of larger datasets like ImageNet computationally prohibitive. Nevertheless, to address the reviewer's comment, we designed additional experiments by finetuning pre-trained models on Stanford Dogs dataset which contains samples from ImageNet. The experiments are being conducted and the results will be released shortly.
>
> **Questions**
>
> **Q1. Why more encoding blocks leads to higher abstraction level? Imagine given enough model parameters, one encoding block can learn the same function as multiple encoding blocks.**
>
> **Answer:** We appreciate the reviewer insight. Our work is focused on practical approaches in diffusion models, where the layers' width are sensible and follow the literature's best practices. We do not observe a commonly adapted implementation of DM happen to replace its deep architecture with one extremely wide hidden layer to approximate its effect, although it is theoretically possible. In the common setups (multi-layered deep network), each layer aggregate its preceding layers' outputs, resulting in handling  more abstract representation.
>
> **Q2. Why more encoding blocks lead to less variance, which increase the model parameters? This seems to contradict with prior works that demonstrate a larger model tends to memorize more (have higher variance).**
>
> **Answer:** As mentioned above, we only focus on well-trained models that are not over- or under-parameterized. Memorization is not caused merely by having a large number of parameters; it arises when the model is over-parameterized relative to the data. A larger model trained on adequately large data may actually show better generalization. In our analysis, we assume and use well-trained models. To corroborate, in Table 3 we reported the cosine similarity and Euclidean distance between generated images and their most similar counterparts (nearest neighbors) in the training data in feature space and pixel space, respectively. It can be seen that our larger model  generations are actually less similar to the training data, which provides a definite proof that the observed high variance is not due to over-fitting (i.e. more number of parameters).

---

> ### Author Response · Authors · 2025-11-23
> **Response to Reviewer KS7X (3/3)**
>
> **Q3. No matter how many encodings blocks a model have, when trained with denoising score matching loss, they are always approximating the data distribution, which in theory should have no bias and variance issue. It is unclear why the observed bias-variance tradeoff emerges.**
>
> **Answer:**  We would like to respectfully emphasize that denoising score matching loss only 'approximates' the data distribution - as the reviewer correctly pointed out. Bias and variance of models manifest due to the 'approximation'. Since the proximity/precision of this approximation is the function of the model hyper-parameters, including its size and encoding blocks etc., the trade-off emerges and gets controlled in the model design. If a model were able to 'perfectly' replicate the complete data distribution irrespective of the number of encoding blocks it uses, then the bias-variance tradeoff would disappear. However, no such model is practicable.
>
> **Q4. Can you do more experiments on dataset with more classes?**
>
> **Answer:** Yes, of course. We are currently conducting experiments on a larger dataset with more classes (Stanford Dogs), and we will report the results soon.
>
> **References:**
>
> [1] C. M. Bishop et al., Pattern recognition and machine learning, volume 4. Springer, 2006.
>
> [2] N. Carlini et al., Extracting training data from diffusion models. USENIX 2023.
>
> [3] T. Cover et al., Nearest neighbor pattern classification. IEEE transactions on information theory, 1967.
>
> [4] J. Ho et al., Denoising diffusion probabilistic models. NeurIPS 2020.
>
> [5] B. Hoover et al., A universal abstraction for hierarchical hopfield networks. Symbiosis of Deep Learning and Differential Equations II, 2022.
>
> [6] B. Hoover et al., Memory in plain sight, NeurIPS 2023.
>
> [7] D. Krotov, A new frontier for hopfield networks. Nature Reviews Physics, 2023.
>
> [8] D. Krotov et al., Large associative memory problem in neurobiology and machine learning. preprint arXiv:2008.06996, 2020.
>
> [9] Y. LeCun et al., A tutorial on energy-based learning. Predicting structured data, 2006.
>
> [10] H. Ramsauer et al., Hopfield networks is all you need. arXiv preprint arXiv:2008.02217, 2020.
>
> [11] R. Rombach et al., High-resolution image synthesis with latent diffusion models. CVPR 2022

---

> ### Author Response · Authors · 2025-11-26
> **Additional Experiment**
>
> In response to Question 4, as an additional experiments on a larger number of classes, we fine-tuned 3 different variations of SD V1.5 on a biased custom dog breed dataset with images from ImageNet. The dataset is customized as follows:
>
> * We took Stanford Dogs with 120 classes of dog breeds taken from ImageNet.
> * Then we introduced bias in the dataset by keeping \(n_c + 20\) samples of each class, where \(n_c\) is the class number. The classes are ordered alphabetically. So, there are 21 up to 140 samples of each class according to their class number.
> * The models are finetuned each for 100 epochs and the learning rate of 1e-8.
>
> The fine-tuned models are:
> * $M_1$: The original model
> * $M_2$: The original model with its U-Net bottleneck layer pruned (less abstract representation)
> * $M_3$: The original model with the same number of bottleneck parameters pruned (97038080), 11.29\% of total model parameters, randomly from the entire U-Net
>
> |                 |               |        |        |        |        |    Class     | Number |        |         |       |        |        |        |
> |:---------------:|:-------------:|:------:|:------:|:------:|:------:|:------:|:------------:|:------:|:------:|:------:|:------:|:------:|:------:|
> |                 | \# of params. |   10   |   20   |   30   |   40   |   50   |      60      |   70   |   80   |   90   |   100  |   110  |   120  |
> |  Complete Model |   859520964   | -4.8\% | -3.1\% | -2.0\% | -2.1\% | -2.3\% |    -2.6\%    | +3.3\% | +3.5\% | +4.6\% | +5.1\% | +5.5\% | +6.7\% |
> |   No Midblock   |   762482884   | -3.9\% | -4.0\% | -1.7\% | -1.3\% | -0.6\% |    +0.1\%    | +4.0\% | +3.4\% | +4.1\% | +4.9\% | +5.4\% | +6.1\% |
> | Randomly Pruned |   762482884   | -4.5\% | -3.7\% | -2.2\% | -2.0\% | -1.5\% |    -1.1\%    | +2.6\% | +3.7\% | +4.6\% | +5.0\% | +5.9\% | +6.4\% |
>
> Using each model, 20000 images were generated with the prompt "a dog". The generated images are then classified using a pretrained dog breed classifier taken from "https://huggingface.co/jhoppanne/Dogs-Breed-Image-Classification-V1". We report, for 10 representative classes, the change in the number of generated images corresponding to each class across the finetuned models. As it can be seen, the model without mid-block layer (less depth and less abstraction) became less biased compared the other two. Furthermore, the randomly pruned model that has the exact same number of parameters the model with no mid-block has almost the same behavior in introducing bias as the complete model with the same depth.

---

> ### Comment · Reviewer_KS7X · 2025-11-27
>
> Thanks for your effort during rebuttal. Currently, I still feel the connection between diffusion models and K-NN is mostly conceptual and kind of vague, hence I am not confident to recommend acception of this work.
>
> The energy model (equation 2) and the related theory makes sense, but cannot directly transfer to diffusion models. Compared to energy model, diffusion models have many unique characteristics, and I don't find it convincing to state the distribution induced by a diffusion model will share similar form as the energy based model. Even if I assume this was true, it is unclear how the core concept, the abstraction level, i.e., temperature $\tau$ is determined. $\tau$ could be determined by many aspects, such as number of sampling steps, type of ODE solver for performing the sampling steps, approximating error of learned the score functions, the optimization process and the architectural bias. I find it not convincing to directly associate $\tau$ with the number of encoding blocks. I will list some specific questions I have as follows.
>
> (i) Number of encoding might correlate with the change of bias-variance tradeoff, but they might not be causally related. It is unclear if it is true that the number of encoding block has casual relationship with bias-variance tradeoff.
>
> Even if they have casual relationship, it is unclear whether number of encoding blocks influences the bias-variance through abstraction level.
>
> The relationship between bias-variance tradeoff, number of encoding blocks and abstraction levels are largely just conjectures.
>
> (ii) Why in diffusion models, the bias-variance tradeoff is associated with the abstraction level? How do you even define this concept in diffusion models?
>
> (iii) There is no evidence the number of encoding block corresponds to the abstraction level $\tau$.
>
> (iv) How does other architecture designs, such as decoders, skip connections influence the bias-variance tradeoff? They don't have any significant effect? Why you only focus on encoders? What about DiT model, which does not have encoders?
>
> (v) Diffusion models have many unique properties that are overlooked by this work. For example, diffusion models are consisted of many score functions corresponding to different noise levels and their properties lead to the bias and variance trade-off. It is important to study how these score functions are related to the abstraction level. When changing the encoder blocks, how does each of the score functions evolve?

---

> ### Author Response · Authors · 2025-12-03
> **Response to Reviewer KS7X (Round 2, 1/2)**
>
> We thank the reviewer for their response. Below, we provide our detailed clarification:
>
> **"The energy model (equation 2) and the related theory makes sense, but cannot directly transfer to diffusion models. Compared to energy model, diffusion models have many unique characteristics, and I don't find it convincing to state the distribution induced by a diffusion model will share similar form as the energy based model. Even if I assume this was true, it is unclear how the core concept, the abstraction level, i.e., temperature is determined. could be determined by many aspects, such as number of sampling steps, type of ODE solver for performing the sampling steps, approximating error of learned the score functions, the optimization process and the architectural bias. I find it not convincing to directly associate with the number of encoding blocks. I will list some specific questions I have as follows."**
>
> **Answer:**
> Thank you for acknowledging that the model and related theory makes sense. We would like to respectfully mention that there  are some obvious shortcomings in the reviewer's understanding.
> We are not concerned about the similarities/differences between diffusion models and energy models. Energy Models are not even directly relevant here. We only consider the `energy landscape' of the underlying representation learned by a model. Such a landscape can be established for any model in general by defining energy with the loss function, e.g., higher loss corresponds to higher energy. We use this for a systematic theoretical treatment of our problem.
>
> From the reviewer's perspective of focusing on the energy landscape as a parallel of energy models, we agree that
>  $\tau$ may get influenced by many factors. However, even under that perspective, it is theoretically demonstrable that abstraction level (through encoding blocks) is a key element influencing the temperature $\tau$. We provide the following proposition with a proof to show that.
>
> **Proposition:** (Higher abstraction levels of representations increase the smoothing factor $\tau$)
>
> Let $x$ be a signal, and let $z = \phi(x)$ denote its  abstract representation. Consider gradient-flow dynamics on $x$ with a smoothing factor $\tau$:
>
> $$\tau \frac{dx}{dt} = - \nabla_x E(x),$$
>
> where E(x) is an energy function.
>
> Then the effective smoothing factor in the abstract space, denoted $\tau'$, is an increasing function of the abstraction level of $z$. That is, higher abstraction corresponds to a larger $\tau'$.
>
> **Proof:**
>
> By the chain rule, the rate of change in the abstract space is
>
> $$\frac{dz}{dt} = \frac{dz}{dx} \frac{dx}{dt}.$$
>
> Taking norms, we have
>
> $$\Big\| \frac{dz}{dt} \Big\| = \Big\| \frac{dz}{dx} \frac{dx}{dt} \Big\| \le \Big\| \frac{dz}{dx} \Big\| \cdot \Big\| \frac{dx}{dt} \Big\|.$$
>
> By Lemma 1 (in the Appendix), the Jacobian norm of the abstraction mapping decreases with higher abstraction:
>
> $$\Big\| \frac{dz}{dx} \Big\| < 1.$$
>
> Hence,
>
> $$\Big\| \frac{dz}{dt} \Big\| < \Big\| \frac{dx}{dt} \Big\|.$$
>
> Since the smoothing factor $\tau$ scales the rate of change along the trajectory, the effective smoothing factor in the abstract space must satisfy
>
> $$|\tau'| > |\tau|.$$
>
> Hence, higher abstraction leads to a larger effective smoothing factor. $\square$
>
> We hope that the above proof concretely clarifies the confusion.
>
> **Questions:**
>
> **Q(i): Number of encoding might correlate with the change of bias-variance tradeoff, but they might not be causally related. It is unclear if it is true that the number of encoding block has casual relationship with bias-variance tradeoff.**
>
> **Even if they have casual relationship, it is unclear whether number of encoding blocks influences the bias-variance through abstraction level.**
>
> **The relationship between bias-variance tradeoff, number of encoding blocks and abstraction levels are largely just conjectures.**
>
> **Answer:** Thank you for the comment. We first theoretically established  the underlying link between abstraction level and bias-variance tradeoff through Theorem 1 and 2, and empirically corroborated the findings with a careful set of experiments that uses abstraction level as the key variable. Our experiments make a practical assumption that all models are well trained and therefore employ sensible architectural design. When a variation in depth (abstraction level) causes a change in model parameter count, we isolate the effect by providing additional results on a model that has comparable parameter size but less encoding blocks. Both theory and experiments throughout our paper are dedicated fully to establish the relationship between bias-variance tradeoff and abstraction levels. We very strongly disagree with the reviewer's remark that does not seem to reflect even a proper reading of the paper, much less reflect on it.

---

> ### Author Response · Authors · 2025-12-03
> **Response to Reviewer KS7X (Round 2, 2/2)**
>
> **Q(ii): Why in diffusion models, the bias-variance tradeoff is associated with the abstraction level? How do you even define this concept in diffusion models?**
>
> **Answer:** We actually dedicated about 6 pages of content (Section 3 of the paper) answering this question. Gxyj rated our presentation 'excellent'. It is unfortunate that this reviewer seems to have no understanding of the foundational idea of concept abstraction.
>
> **Q(iii): There is no evidence the number of encoding block corresponds to the abstraction level $\tau$.**
>
> **Answer:** It is a fundamental principle in deep learning that deeper encoding layers produce progressively more abstract representations of the data [1].
>
> **Q(iv): How does other architecture designs, such as decoders, skip connections influence the bias-variance tradeoff? They don't have any significant effect? Why you only focus on encoders? What about DiT model, which does not have encoders?**
>
> **Answer:** We refer to abstraction levels independently of the specific model architecture. Our theoretical results encompass any design choice that contributes to concept abstraction. Evaluation of what choices contribute to concept abstraction is not within the scope of our paper as that is an entirely different question. Our empirical evaluation isolates the most well-established design choice of the number of (encoder) layers/blocks (in diffusion context) impacting abstraction, and explores it thoroughly to corroborated the theoretical results.
>
> Regarding DiT-style architectures, even in that case, abstraction naturally emerges. Transformer layers inherently form a progression of abstraction through their global receptive field and iterative mixing of information. Self-attention allows each token to integrate information from the entire spatial domain—often capturing broader context than the strictly local receptive fields of convolutional layers in U-Nets [2]. As depth increases, tokens aggregate information from increasingly large and diverse context sets, yielding higher-level, more invariant, and semantically richer representations. This structured evolution of representations therefore plays the same functional role as the increasing abstraction seen in hierarchical encoder–decoder architectures.
>
> **Q(v): Diffusion models have many unique properties that are overlooked by this work. For example, diffusion models are consisted of many score functions corresponding to different noise levels and their properties lead to the bias and variance trade-off. It is important to study how these score functions are related to the abstraction level. When changing the encoder blocks, how does each of the score functions evolve?**
>
> What the reviewer suggests, is an entirely different direction of exploration. We are not concerned about the latent space associated with noised images across timesteps. While different score functions and noise levels may (or may not)  influence the bias–variance tradeoff, our analysis is about establishing a link between this trade-off and the abstraction level. Whereas we do not argue about the importance of further studies for understanding the bias-variance tradeoff in diffusion models in general, the suggested study is entirely beyond the scope of our paper.
>
> **References:**
>
> [1] Y. LeCun, Y. Bengio, and G. Hinton. Deep learning. nature, 521(7553):436–444, 2015.
>
> [2] M. Raghu, et al., Do vision transformers see like convolutional neural networks? NeurIPS 2021.

---

### Author Response · Authors · 2025-12-03

We thank the AC for their valuable time. For the AC's convenience, here we provide an up-to-date summary of our responses to the reviewers.

We submit this work to a leading machine learning venue owing to its conceptual depth, novelty and intriguing findings. We generally received very encouraging feedback from the reviewers, who acknowledged our work as **novel** (Gxyj), **clearly organized, interesting and new** (rj1q), **interesting** and **inspiring further investigation** (KS7X).

We submitted our responses to the reviewers on Nov 23, promising adding further results shortly, which were added on Nov 26.


On Nov 27, Gxyj confirmed that all their concerns were well addressed and confirmed the score of 8 - verifiable in their final comment.

Reviewer rj1q could not reply to our rebuttal before the sudden shutdown of the reviewer input after OpenReview leak. However, as can be seen, their noted Weaknesses are mostly minor presentation and discussion related concerns. We have already addressed all of those, and provided concrete mathematical explanations where required.

On Nov 28, KS7X asked further questions, which we have answered. However, we would like to point out some critical issues about this reviewer. Whereas we respectfully replied to the reviewer, their comments  do not reflect an informed review in good faith. The reviewer challenged and disregarded well-established foundational knowledge in the field at multiple occasions, and provided a largely senseless feedback. We quote some of their questions below to exemplify:

- The reviewer says: "Question 1: *Why more encoding blocks leads to higher abstraction level? Imagine given enough model parameters, one encoding block can learn the same function as multiple encoding blocks."*

Its a well-established phenomenon in deep learning that in a typical sensible architecture, more levels of encoding leads to higher abstraction levels of the concepts. Reviewer's followup  remark on replacing the encoder with a single block either reflects their extremely poor grasp of the foundations of deep learning or it is a deliberate attempt to senselessly belittle our work.

- The reviewer says "Question 3: *No matter how many encodings blocks a model have, when trained with denoising score matching loss, they are always approximating the data distribution, which in theory should have no bias and variance issue."*

This comment is absurd. It implies that training any model (independent of the underlying network architecture) using denoising score matching loss always guarantees a perfect model of the data distribution. This reasoning challenges the whole foundation of current developments in diffusion modeling.


- Reviewer says "Question i (post rebuttal): *There is no evidence the number of encoding block corresponds to the abstraction level"*

This is again a clear indication of disregarding the basic knowledge in the field.

Whereas we have still responded to KS7X respectfully with further concrete replies, we found it imperative to highlight the issue to the AC, and  we hope that the AC will either rescind KS7X's review or carefully gauge their feedback on our work. We strongly believe that our paper presents intriguing new results in an important research direction, thoroughly backed by theoretical and empirical evidence and ICLR is the best avenue to disseminate these potentially impactful results.

Kind regards,

Authors.

---

### Meta-Review · Area_Chair_mUP2 · 2025-12-06

**Summary:**

This paper investigates a bias, i.e., variance tradeoff in diffusion models. The authors propose that the abstraction level of intermediate representations, which is operationalized primarily via the number of encoder blocks, controls the smoothness of the model’s implied energy landscape. High abstraction yields smoother landscapes that amplify dataset bias, while low abstraction yields more localized minima that cause memorization and privacy risks. The paper combines theoretical arguments with empirical evaluations on MNIST, CelebA, and Stable Diffusion v1.5.

The reviewers agree that the core idea that connects bias amplification and memorization through a variance perspective is intriguing. The empirical observations appear consistent with this viewpoint. However, significant concerns were raised regarding the theoretical grounding, particularly the applicability of the proposed energy-landscape analysis to diffusion models. The definition of "abstraction level" is unclear. The observed correlation between encoder depth and bias–variance behavior might not demonstrate a causal mechanism. The assumptions used in the theoretical results might not hold for diffusion models. Experimental scope remains limited, where the overall empirical evaluation relies heavily on small datasets.

The rebuttal successfully addressed some concerns. For example, the authors clarified notation issues, provided more explicit definitions of abstraction and energy landscapes, and added new experiments on Stanford Dogs to strengthen the empirical evidence. However, the key conceptual concerns raised by reviewers KS7X and rj1q remain largely unaddressed, including the lack of a rigorous, diffusion-model-specific definition of abstraction; the insufficient justification that encoder depth controls an effective temperature; the speculative nature of the causal claims; and the limited connection between the theoretical framework and the actual mechanics of diffusion sampling.

Since the theoretical foundation is a core claim of the paper, the concerns from most reviewers are valid. The authors are encouraged to tackle these issues in a future revision.

**Reviewer Concerns:**

Concerns Addressed:

1. Notation inconsistencies were acknowledged and corrected. (Reviewer rj1q)
2. New Stanford Dogs experiments partially addressed concerns on limited datasets. (Reviewer KS7X, Gxyj)
3. Provided explanations regarding abstraction in DiT-style architectures. (Reviewer Gxyj)

Remaining Concerns:

1. The connection between diffusion models and the proposed energy-model analogy remains vague and largely conceptual. (Reviewer KS7X)
2. The theory remains general to neural networks and is not tailored to diffusion processes. The reviewer’s request for diffusion-model-specific formulations remains. (Reviewer rj1q)
3. Lack of evidence that encoder depth causally determines an abstraction-related "temperature" parameter. (Reviewer KS7X, rj1q)
4. The definition of abstraction is still unclear. No model-specific or measurable quantity was provided. (Reviewer KS7X)

**Reviewer Scores:**

Reviewer KS7X: Score would remain 2 (reject). The reviewer explicitly stated that the rebuttal did not resolve their core conceptual concerns.

Reviewer rj1q: Likely remains at 4 (marginally below acceptance). Their main theoretical concerns remain unaddressed.

Reviewer Gxyj: Maintains 8 (accept) after rebuttal and explicitly confirmed their satisfaction.

---

### Decision · Program_Chairs · 2026-01-26

Reject